# 'The broker also told me that I will not have problems after selling because we have two and we can survive on one kidney': Findings from an ethnographic study of a village with one kidney in Central Nepal

**Bijaya Shrestha**[1]*, **Bipin Adhikari**[2,3], **Manash Shrestha**[4], **Ankit Poudel**[5], **Binita Shrestha**[6], **Dev Ram Sunuwar**[7], **Shiva Raj Mishra**[1,8], **Luechai Sringernyuang**[4,9]

1 Academy for Data Sciences and Global Health, Kathmandu, Nepal, 2 Mahidol-Oxford Tropical Medicine Research Unit, Faculty of Tropical Medicine, Mahidol University, Bangkok, Thailand, 3 Centre for Tropical Medicine and Global Health, Nuffield Department of Medicine, University of Oxford, Oxford, United Kingdom, 4 Department of Society and Health, Faculty of Social Sciences and Humanities, Mahidol University, Nakhon Pathom, Thailand, 5 Independent Researcher, Bharatpur-05, Chitwan, Nepal, 6 Independent Researcher, Tanahun-05, Nepal, 7 Department of Nutrition and Dietetics, Armed Police Force Hospital, Kathmandu, Nepal, 8 Melbourne School of Population and Global Health, University of Melbourne, Melbourne, Australia, 9 Contemplative Education Center, Mahidol University, Nakhon Pathom, Thailand

* bijayashresthanepal@gmail.com

## Abstract

Kidney selling is a global phenomenon engraved by poverty and governance in low-income countries with the higher-income countries functioning as recipients and the lower-income countries as donors. Over the years, an increasing number of residents in a village near the capital city of Nepal have sold their kidneys. This study aims to explore the drivers of kidney selling and its consequences using ethnographic methods and multi-stakeholder consultations. An ethnographic approach was used in which the researcher lived and observed the residents' life and carried out formal and informal interactions including in-depth interviews with key informants, community members and kidney sellers in Hokse village, Kavrepalanchok district. Participants in the village were interacted by researchers who resided in the village. In addition, remote interviews were conducted with multiple relevant stakeholders at various levels that included legal workers, government officers, non-government organization (NGO) workers, medical professionals, and policymaker. All formal interviews were audio-recorded for transcription in addition to field notes and underwent thematic analysis. The study identified processes, mechanisms, and drivers of kidney selling. Historically, diversion of a major highway from the village to another village was found to impact the livelihood, economy and access to the urban centres, ultimately increasing poverty and vulnerability for kidney selling. Existing and augmented deprivation of employment opportunities were shown to foster emigration of villagers to India, where they ultimately succumbed to brokers associated with kidney selling. Population in the village also maintained social cohesion through commune living, social conformity (that had a high impact on decision making), including behaviours that deepened their poverty. Behaviours such as alcoholism, trusting

**Data Availability Statement:** Data cannot be shared publicly because of the nature of the data being qualitative that contains personal quotes and clues to where the study occurred and can be potentially identifiable. However, data is available on request to the chair of the Academy for Data Sciences and Global Health (ADSGH), Kathmandu, Nepal (E-mail: shivarajmishra@gmail.com) complying with the data access policy by Institutional Review Committee (IRC) of Nepal Health Research Council and ADSGH.

**Funding:** The authors received no specific funding for this work.

**Competing interests:** The authors have declared that no competing interests exist.

and following brokers based on the persuasion and decision of their peers, relatives, and neighbours who became the new member of the kidney brokerage also contributed to kidney selling. The other reasons that may have influenced high kidney selling were perceived to be a poor level of education, high demands of kidneys in the market and an easy source of cash through selling. In Hokse village, kidney selling stemmed from the interaction between the brokers and community members' vulnerability (poverty and ignorance), mainly as the brokers raised false hopes of palliating the vulnerability. The decision-making of the villagers was influenced heavily by fellow kidney sellers, some of whom later joined the network of kidney brokers. Although sustained support in livelihood, development, and education are essential, an expanding network and influence of kidney brokers require urgent restrictive actions by the legal authority.

## Background

### Kidney selling and its epidemiological burden

Globally, 150,000 kidneys are transplanted annually, of which the majority arrived through the kidney selling [1]. The volume of kidney selling has increased steadily over the years—accumulating $US1.7 billion transaction annually [2, 3]. Kidney selling is associated with a growing burden of End-Stage Renal Diseases (ESRD) globally. Of those with ESRD, nearly 2.5 million people (projected to double to 5.4 million by 2030) are currently in renal replacement therapy (RRT) and another 2.3–7.1 die prematurely due to lack of RRT and transplantation [4]. The growing burden of ESRD and demand for transplantation (a feasible alternative against long term dialysis) has fueled kidney selling in low-income countries—orchestrated often through an informal network by the brokers [5].

### Global scenario of kidney selling

With an exception of Iran, kidney selling is illegal around the world [6]. The phenomenon of kidney selling was first tracked by Nancy Scheper-Hughes during her field visit to the shantytowns of Brazil, which she framed as 'body-snatching rumours' [7] and her continued investigations in South America, Africa, Europe, and Asia [8]. Kidney selling is prevalent in countries such as Brazil, China, Moldova, the Philippines, Romania, Turkey, Egypt, Bolivia, and Peru, while the buyers come from high-income countries like Saudi Arabia, Israel, Oman, Japan, Australia, United States, Canada, and Italy [9].

In South Asia, the Indian subcontinent (e.g., India, Bangladesh, and Pakistan) in particular is a prominent hotspot for kidney selling. In these countries, brokers often persuade people to sell their kidneys through false information about the bodily function of kidneys and by enticing them with a considerable sum of financial incentive [8–11]. Countries in the Indian subcontinent are therefore considered as the 'kidney hub' for the rich westerners [9, 12].

### Potential reasons for kidney selling in the Indian subcontinent

Among several reasons, poverty is a significant contributor that triggers kidney selling in the Indian subcontinent [9]. In his 2001 article, anthropologist Lawrence Cohen narrated the stories of poor Indian slum dwellers who sold their organs to raise money for the dowry of their daughters [13]. Echoing India, countries such as Pakistan and Bangladesh have also emerged as safe places for kidney transplantation. Poor people are drawn into kidney selling in the

dreams of overcoming their poverty, debt and sometimes kidney donation (selling) is framed as an act of moral obligation and altruism referring to religious beliefs [14–18]. Furthermore, the unregulated medical industry and lackadaisical regulatory measures in these low- and middle-income countries have created a favorable environment for the kidney selling [15, 19].

### Context of kidney selling in Nepal

Nepal is a small country sandwiched between two giant neighbors, China in the North and India elsewhere. The federal democratic republic of Nepal is divided into seven provinces. Each province has eight to 14 districts. The districts make up the local administrative units referred to as urban and rural municipalities. There are 753 local units, including six metropolitan municipalities, 11 sub-metropolitan municipalities, 276 municipalities, and 460 rural municipalities [20]. Nepal remains one of the most volatile areas for political instability with a decade-long Maoist civil war (1996–2006) and frequent political interferences [21]. In addition, unemployment, lack of infrastructure development, caste hierarchy, and illiteracy make Nepalese people vulnerable to cheap labor and organ trafficking [22, 23]. To compound the situation, the porous border between Nepal and India have facilitated many Nepalese people in succumbing to the organ trafficking [24, 25].

Hokse, a small village located in the Kavrepalanchok district situated 50 km northeast of Kathmandu, has been established as a hub for kidney selling. Over the past decades, nearly a third of the villagers have travelled to India to sell their organs. The village is infamous for being one of the cheapest sources for kidney buyers and is being stigmatized as a 'kidney village' in Nepal [9]. Previous accounts of kidney sellers are filled with deception by the brokers, false expectations, medical complications, and adverse consequences. The monetary incentives of kidney selling also seem to be driven by ultimate purchase capacity to buy phones, televisions, motorbikes, houses, and land Fields [26].

This research explored the reasons behind kidney selling in Hokse village, using ethnographic methods. Although kidney selling is a global phenomenon, Hokse village bears a unique social, cultural, and demographic characteristic, making it critical to explore the complex interplay of the plethora of factors behind concentrated kidney selling in the village.

## Materials and methods

### Setting

The study was carried out in the Hokse village, ward numbers 7 and 8 of Paanchkhal Municipality, Kavrepalanchok district (Fig 1). The village is about two hours mountainous drive (50 kilometers) from the capital Kathmandu to the east. People from different ethnicities such as *Bahun*, *Chettri*, *Tamang*, *Sarki*, and *Danwar* live in a total of 1000 households in these wards, and their livelihoods depend primarily on subsistence farming and daily wage-earning [27].

### Study design and participants

A detailed study protocol of this research has been published elsewhere [28]. Briefly, this study utilized an ethnographic approach to understand how local social and cultural factors have affected kidney selling in the village. This study follows a standard qualitative methodology outlined by a COREQ checklist (S1 File). In-depth interviews were conducted with the primary participants who were kidney sellers aged above 18 years (Table 1). We also carried out non-participant observations and informal interactions with community members who were permanent residents of the village. In addition, we interviewed relevant medical professionals, policymakers at the ministry of health, legal workers, and non-governmental organization

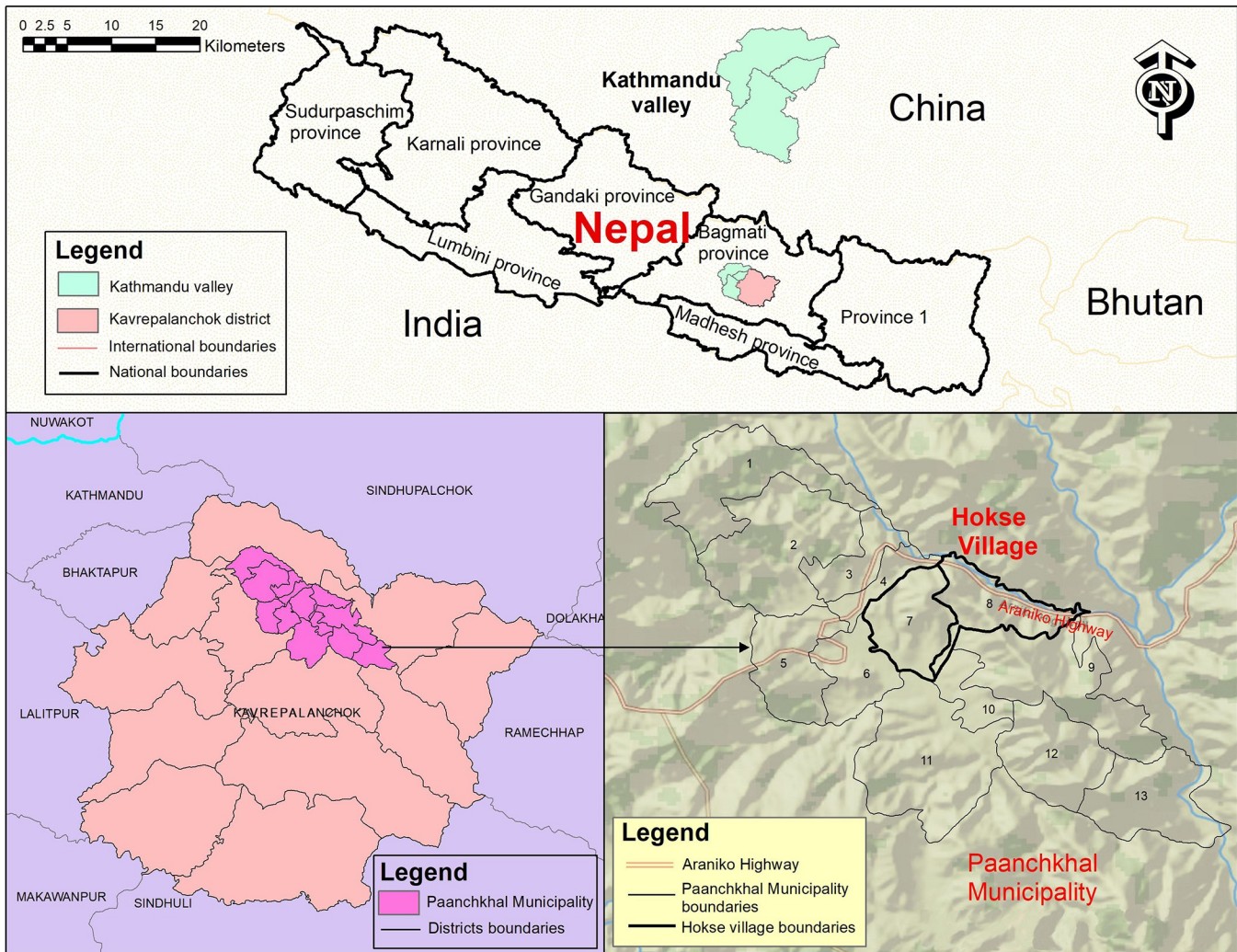

**Fig 1. The study site is selected Hokse village, ward numbers 7 and 8 of Paanchkhal Municipality, Kavrepalanchok district, Nepal.** The map was created using ArcGIS desktop version 10.8. The shapefile of the administrative districts and location for Nepal was obtained from the Government of Nepal, Ministry of Land Management, Survey Department website and were publicly available for unrestricted use (http://www.dos.gov.np/nepal-map).

(NGO) staff working against organ and human trafficking for at least one year as key informants using online meeting platforms. We also interviewed two border patrol officers who were deployed at a vital border checkpoint for at least six months.

Altruistic kidney donors were not included in this research. Informed consent was first obtained from the participants verbally. After sharing the study information, the participants

**Table 1. Details of the study participants.**

| Type of participant | Study population | Number |
|---|---|---|
| Primary participant | Kidney sellers of *Hokse* village | 10 |
| Key informants | Family members, neighbours, relatives | 15 |
| | Health care providers and local governmental officers | 6 |
| Relevant stakeholders | Transplant Unit's medical personnel | 2 |
| | Policymakers, legal workers, NGO/INGO workers | 8 |
| | Border checkpoint officers | 2 |

were requested to sign the consent form. As this research was conducted using an ethnographic approach, the researcher familiarized himself with the community members by living in the village, observing and also participating in the village's culture and tradition. We obtained informed consent for all formal interviews. In the case of online interviews, we obtained verbal consent from the participants.

Primary participants and key informants were asked to choose the interview site and if they would like to be interviewed or not. Respondents were assured that they could discontinue the interview at any point in time without requiring to provide any justifications. All participants were anonymized in the collected data and all personal identifiers were avoided from the transcripts.

The participants in this study is composed of a diverse group of respondents, which allowed identifying and collecting information-rich cases for an in-depth study of our topic [29]. Furthermore, as the issues around kidney trade can be culturally sensitive, we used a snowball approach to identify additional participants till we reached the data saturation [30]. However, the sampling design also evolved during the study period based on the circumstances of data collection, opportunities to enrol relevant participants and capture emerging issues.

## Data collection

The principal investigator (BS[1]) and co-investigator (AP) conducted the ethnographic fieldwork. The fieldwork started in February 2021 and the data analysis was completed in September 2021. BS[1] conducted initial fieldwork to build a relationship with the villagers and form an initial understanding of the social and cultural fabric of the village. BS[1] and AP visited subsequently and stayed in the village for various periods of duration under the supervision of BA. BA has extensive research experience around ethnography and qualitative research and was based in Nepal during the data collection period. AP and BS[1] collaborated in the previous research as well and all study documents (proposal, consent form, ethical approval forms from Institutional Review Board, and the interview guide) were shared. AP and BS[1] interacted regularly to share their interpretations, similarities and differences and sought suggestions and external interpretations from BA and the rest of the authors. Abiding by the national restrictions during the COVID-19 pandemic, BS[1] conducted remote key informant interviews (KII) with different stakeholders using the online Zoom meeting platform. BS[2] coordinated the online meeting with the KII.

Formal interviews were carried out using a topic guide to collect data. Each interview lasted from 30 minutes to two hours. Multiple in-depth interviews (IDI) were conducted with kidney sellers to understand their story and background status. A field diary was used to record important details during observation such as non-verbal communication, including the personal presentation of the participant, body expressions, gestures, facial expressions, style, and alterations in speech (silences, choking speech, blatant speech, fading speech, cringing and tremors), laughter, and other manifestations.

BS has an educational background in health and social sciences and belongs to a *Newar* ethnic community in western Nepal where other ethnic groups such as *Bahun*, *Chettri*, *Magar*, *Gurung*, *Darai*, and *Bote* also reside, similar to that of Hokse village [31]. AP is a graduate of Business Administration and is familiar with the local social, political, and cultural context of Nepal. Other team members included an expert medical anthropologist (LS), a clinician and social scientist with expertise in community engagement (BA), a socio-demographer familiar with the local context (DRS), a public health expert with extensive experience in non-communicable diseases including kidney diseases (SRM), and a public health professional with experience in gender-related research (MS).

## Data analysis

The audio-recorded interviews and field notes were transcribed first in the Nepalese language and then into English retaining the nuanced meanings and interpretations during translation. The interview data were sorted and labelled, followed by reiterative reading, and understanding, initial coding and theme building. Finally, the data were analyzed and interpreted using the multilevel theoretical framework of Baer, Singer and Johnsen's Critical Medical Anthropology [32].

Content analysis was conducted independently by two investigators (BS and BA). The initial results were shared with LS, who supervised refining the codes and themes. NVivo version 10 (QSR International, Melbourne, Vic., Australia) was used to manage and analyze the data. The analysis process and findings were shared with the research team. Any disagreements between the theme during the analysis were discussed and resolved after seeking an alternative opinion from the other team member.

## Ethical approval

The study protocol was reviewed and approved by Mahidol University Central Institutional Review Board (MU-CIRB 2020/217.1808), Thailand, and the Nepal Health Research Council (716/2020 PhD), Nepal.

## Results

### Participant characteristics

A total of 43 participants were enrolled in this study, including nine kidney sellers and one kidney seller who was also a broker (Tables 1 and 2). In the village, most of the kidney selling occurred between early 1990s and 2000, the kidney sellers interviewed in this study were in their mid-20s to 30s when they sold their kidneys. Higher proportion of males were engaged in kidney selling compared to females (Male versus female: 7:3) and it accurately reflects the kidney sellers' population in the village.

One of the kidneys seller who later became broker was working as a chef in an Indian restaurant who realized an opportunity to make an extra money by exploiting the familiarity and trust he had in his own village. Secondary participants comprised of key informants such as family members, neighbors, and relatives of the kidney sellers (n = 15) and local healthcare providers and governmental officers (n = 6). Two border checkpoint officers were also

**Table 2. General characteristics of kidney sellers.**

| S. No. | Age | Gender | Caste | Employment status |
|---|---|---|---|---|
| 1 | 58 years# | Male | *Bahun* | 1st kidney broker from the village |
| 2 | 60 years | Male | *Newar* | Business |
| 3 | Early 40s | Female | *Mijars* | Daily wage worker |
| 4 | 45 years | Male | *Mijars* | Pig rearing and agriculture |
| 5 | 53 years | Male | *Tamang* | Daily wage worker |
| 6 | 48 years | Female | *Tamang* | Daily wage worker and local alcohol pub |
| 7 | 52 years | Male | *Bahun* | Daily wage worker/irregular |
| 8 | 48 years | Female | *Tamang* | Daily wage worker |
| 9 | 50 years | Male | *Bahun* | Daily wage worker |
| 10 | 52 years | Male | *Bahun* | Daily wage worker /irregular |

#Kidney sellers who became broker later.

included in the interview. Relevant stakeholders were enrolled for remote interviews (n = 10) as indicated in Tables 1 and 3.

## Drivers of kidney selling in Hokse

Kidney selling in Hokse village was found to be a product of an interaction between community members' vulnerability and contextual factors, with the brokers playing a central role (Fig 2). Below, we present our findings on how vulnerability and contextual factors contributed to the concentrated kidney selling phenomenon in the village.

## Vulnerability

Multitude of factors contributed to the vulnerability of the villagers for kidney selling that ranged from characteristics inherent in an individual to wider social factors.

**Poor socio-economic status.** Poor socioeconomic status was a major factor for kidney selling in the village. Conversations with the community members highlighted that unemployment in the village gave little options to support their daily livelihood. As a result, people often

**Table 3. General characteristics of Key Informants.**

| Key informant | Details |
| --- | --- |
| **1. Key informants from the village** | |
| **a. Family members, neighbours, relatives (15)** | FCHV-1 |
| | FCHV-2 |
| | Principal |
| | Treasurer of Ward |
| | Local News reporter |
| | Sellers's mum |
| | School teacher |
| | Local activist |
| | Local priest |
| | Local people shifted to Tamaghat |
| | Kidney sellers' relatives—1 |
| | Kidney sellers' relatives—2 |
| | Kidney sellers' neighbor -1 |
| | Kidney sellers' neighbor -2 |
| | Local person shifted to Tamaghat |
| **2. Stakeholders** | |
| **a. Transplant Unit's medical personals (2)** | Transplant surgeon (10 years of working experience) |
| | Transplant nurse (6 years of working experience) |
| **b. Policymakers, legal workers, NGO/INGO workers (8)** | Policy maker (former Nepal Planning Commission) |
| | Lawyer-legal worker |
| | Police officer- posted in the *Panchkhal* Municpality and registered 1st case |
| | Government officer- Human Organ transplant center* |
| | NGO-1 |
| | NGO-2 |
| | NGO-3 |
| | NGO-4 |
| **c. Border checkpoint officers (2)** | NGO worker |
| | Government officer |

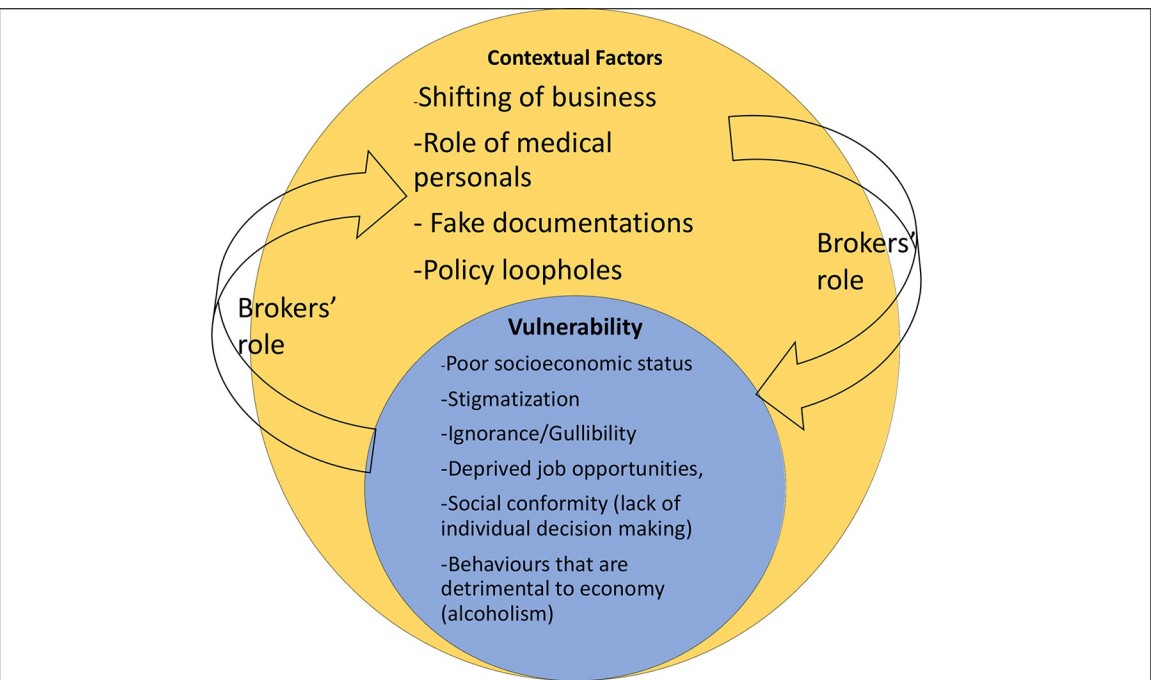

**Fig 2. This figure outlines our broad category of themes: Vulnerability and contextual factors for kidney selling and the critical role of brokers.**

travelled to Kathmandu or India for work. Most of the community members who travelled to India ended up taking unskilled works such as construction labor, security guard, and cleaner at a restaurant. Poverty and a lack of job opportunities meant that any unexpected sum of money for a kidney was alluring enough to overlook any health issues, as one participant pointed out:

*Poverty is one of the major reasons for kidney selling. . . . . . . . . .and no one was aware regarding what would happen after selling their body parts. No one informed the villagers against selling their kidneys. . . . . . . . .. They were tempted by the promised money as well.*

*(58 years old, male, a school teacher from the village, IC)*

*I had never thought of that amount of money in my life. I thought if I get that amount of money in my life, I don't have to work anymore and can enjoy my life. . . . . .. So, I decided to sell it. I made my decision myself.*

*(52 years old, Bahun, male, kidney seller, IDI)*

*I went to India and sold my kidney, and I got the money. . . . . . . . . . . .If it is going to be good for someone, then it is not bad to give a kidney. I got around NRS 125,000 and I have no regrets about it. I made my house after selling my kidney, but the earthquake of 2015 destroyed it.*

*(48 years old, Tamang, female, kidney seller, IDI)*

In contrast, other community members were sharply critical of the kidney sellers in the village, mostly as the kidney sellers spent their money drinking alcohol. They were also disapproving of the kidney selling as the village gained substantial attention from the national and

international media for being notorious for cheap kidneys followed by discrimination and stigma to the entire village. Some respondents questioned if poverty should be considered as a sole cause of kidney selling:

> *Why do you need to go and talk to kidney sellers in **that** kidney selling village? They have embarrassed everybody. They sold their kidneys to drink alcohol, what else? If you think poverty is the reason to sell their kidney, then I am poorer than them. I am still working for my life... They are lazy and want to drink, that's it. Poverty is not the reason......*

> *(57 years old, male, poultry businessman, IC)*

**Alcohol drinking.** Some community members were found drunk as early as 11:00 hours. Visiting local pubs known as *"Bhatti"* in Nepali, for drinking at irregular times such as early in the morning was an accepted local norm. During these situations, many kidney sellers reported meeting the brokers who bought them drinks and food. These brokers used the *"Bhatti"* as a venue to lure innocent community members, presenting them with possibilities of job opportunities in India, and ultimately, kidney selling.

> *Drinking was my habit and we met different kinds of people while drinking. We had met that broker in the local pub. I was influenced by his words and actions. After a few visits with him, I travelled with him to sell my kidney. I did not think so much then. I just went with my friend and sold it. There were some other Nepalese with me in Uttar Pradesh, India, whom I met when I reached there. I did not know them.*

> *(45 years old, Mijar, male, kidney seller, IDI)*

**Ignorance/Gullibility.** People from the village were easily enticed by the brokers using stories of financial compensations after kidney removal. Many brokers formed false kinships and convinced sellers to sell their kidneys. Community members were gullible and got deceived by this kinship and the amount of money they were promised by the organ receiver families. One drunk kidney seller revealed:

> *You might say I am drunk, but I don't mind. Certainly, I was trapped in this case. He [the broker] said that the kidney-receiving family had around 30 to 40 ropanies* of land where there is an irrigation facility and promised me that the family will share the land with me as one of their family members. I don't know if the deception was intentional or not. I was taken to Madras, Santosh hospital. The operation took place at around 2 PM. They gave me two tablets and I was unconscious. After the operation, I wanted to meet the family to whom I had donated. But I was sent back to Kathmandu, directly via the Sunauli border without getting a chance to meet them. I don't know if the kidney receiver is dead or alive, neither do I know where that family's home is.*

> *\* 1 Ropani land = 5476 Sq. feet*

> *(52 years old, Bahun, male, kidney seller, IDI)*

**Social conformity.** People in the village had been victim to a unique trend of kidney selling, particularly because of a tendency to follow peers, neighbors or other community

members, and this was often shown to be due to high communal reliance and trust among the community members. This was also evident in community members' decision to work in India—often they followed to look for opportunities in India rather than cities in Nepal. They had learned that people earned more money by selling their body parts.

> *Many of our community members have already sold their kidneys, and I followed them. Nobody asked any questions nor bothered to discuss as everyone was doing it.*
>
> *(40 years old, Tamang, female, kidney seller, IDI)*

Moreover, kidney selling thrived once the higher caste people such as *Bahun* and *Chettri* joined in. This was like a symbolic gesture to all the people that kidney selling was normal. Eventually, many community members who belonged to lower caste and were considered naive *'Sojo'*, simply followed the actions of their higher-caste peers.

> *The sellers are Bahun and it slowly moved to Dalit and Tamangs. When the higher caste people sold their kidneys, the low caste started to think that it is ok to sell their kidneys as it was approved by higher caste people……..*
>
> *(40 years old, female NGO worker, KII)*

**Indirect incentives.** Many community members explained that kidney selling was indirectly promoted by financial and material incentives offered by various organizations supporting 'struggling kidney sellers.' Unfortunately, financial help from these well-intended organizations was misinterpreted and even exploited to an extent that it promoted the selling of kidneys. Many community members reported that their neighbors sold their kidneys to gain financial incentives and receive aid such as goats, water tanks, and direct cash from the supporting organizations.

> *There was an American doctor who helped here financially. She gave goats to these people at her expense. She did all these with good intentions. Some people protested by saying that this will encourage other people to sell kidneys. People will start thinking that if they sell a kidney then they will get a donation.*
>
> *(55 years old male, Ward chairman of the village, KII)*

## Contextual factors

Multitude of external factors in and outside the village affected as contextual influences that provided the background and aided in promoting the villagers into kidney selling:

**Shifting of business.** The development of a new motorway–*Araniko* Highway, through *Tamaghat* changed the course of once-flourishing Hokse *village*. This shift of the highway away from Hokse village was followed by a subsequent shift in the local business center to Tamaghat, reducing the economic activities of the village. This situation led to a movement of people to Tamaghat and unemployment in the village. Community members in Hokse village then had to resort to work as cheap wage laborers. Furthermore, after their seasonal agriculture, people started travelling to Kathmandu or India for job opportunities. This shifting of local economic activities made the villagers more vulnerable to kidney selling.

*This place used to be a center for people traveling from Kathmandu and Tibet. This was a buzzing area but ever since the construction of Araniko Highway, the business has shifted to Thamaghat and also the people.*

*(63 years old, male, former school teacher, KII)*

**Role of medical personnel.** The role of medics in the kidney selling phenomenon was also critical. Although medical personnel understood that a lot of kidney selling was being orchestrated by brokers through forged official documentation, they had little to almost no options but to accept when they were approached for kidney transplant operation.

*We as medical people have to operate if the documents are well prepared. In some cases, it is so easy to get documents. They prepare the document within 24 hours. I have operated in those circumstances. People with higher authority call and I have no option but to operate. We confirm the documents, and it is perfect so we do not have any options but to perform our duty.*

*(42 years old, male, transplant surgeon, KII)*

Most authorities knew the situation of doctors and donors, but they were obliged by the legal documents produced. There were adequate legal loopholes in which kidney selling occurred. Doctors could suspect the sellers through their appearance, dress-up, follow-up arrangement, and various other clues. Nonetheless, they were constrained by the seller's signature in the consent form.

*Doctors are the main culprits. They know everything but still, they do the operation. For example, a British army guy having a Mongolian face brings an Aaryan-faced donor as their family member. . ...can't that doctor identify [suspect] that there is something wrong?*

*(40 years old, male, police officer, KII)*

*The doctors did not say anything. They gave me medicine for 7 days. The person to whom I donated, had to take medicine every day but I had to take it only till I stayed at the hospital. Doctors told me to drink more water and don't drink alcohol . . . . . .. ..[the laughter indicating that he did not stop drinking alcohol after the operation]*

*(52 years old, Bahun, male, kidney seller, IDI)*

**Fake documentations.** Easy availability of fake documents and consents are another major legal loopholes that promoted kidney selling. The transplantation act permitted altruistic organ donation and brokers were able to produce an authentic-looking document that made the kidney sellers look like altruistic donors. The documents were prepared so precisely that there were no spaces for questions both legally and for medical practice.

*Many high-level society people come and request kidney transplantation. They can prepare the required documents within 24 hours. We did not have any options but to operate as they presented all the required documents, and the documents are legal and certified. Even if we know and suspect that someone may not be entirely fit for the kidney transplantation, they have the documents, and we have no options.*

*(42 years old, male, transplant surgeon, KII)*

*The documents are prepared by the brokers and they are as perfect as originals. We are not able to find the differences between the originals and fake documents. The brokers are equipped with original stamps, logos and all the necessary elements required for producing the documents.*

*(42 years old, male, transplant surgeon, KII)*

**Policy loopholes.** Inadequate awareness on human organ transplantation and a porous border between India and Nepal provided fertile ground for kidney trade. Kidney transplantations also involve a lot of patients at the end-stage of the disease such as brain death, the policies for which are vague in Nepal. Importantly, the policy fails to address the moral dilemma patients' parties might have when seeking such consent. So, practically, seeking written informed consent from patients in critical conditions is more difficult than outlined in the law.

*In Nepal, it is not easy to get brain dead as kidney donors. Patient's attendees start to have negative connotations regarding the people who request a donation as the patient is still in bed and not yet declared dead. We have to prepare proper laws regarding brain death cases and organ transplants in Nepal.*

*(42 years old, male, transplant surgeon, KII)*

There are both government and non-governmental organisations working at the border between Nepal and India. One of such organizations was the Immigration Department of Nepal, which maintains all legal documents including travel documents for foreigners travelling across the border. Border officials (immigration officials and police officers) were unaware of illegal kidney selling but stressed that they were aware of human (mostly women) trafficking.

*I am not aware of these issues here. I have heard more about women trafficking. There are some organizations here working against women trafficking. Our work here is to issue a visa for foreigners. But for the Indian and Nepalese, we do not check anything. It is an open border, so we don't have to do anything.*

*(55 years old, male, government worker, KII)*

The open Nepal-India border does not require any documents from Nepalese citizens and makes it easier to cross the border. Recognizing and stopping potential kidney sellers at the border point was not possible as the kidney selling could only be identified after mutilation of body parts.

*How can we know exactly about kidney sellers? It is not easy to track all the people walking through this border. We can only help in the case of women trafficking but in the case of kidney trafficking, it is not possible.*

*(55 years old, female, NGO worker at the border, KII)*

Kidney selling was only reported to the authorities when there were problems in a financial settlement. Given the high number of people cross the border without documentation,

authorities explained the difficulties of spotting the illegal human trafficking; and organ trafficking was way more covert for them to identify at a border crossing.

*The identification of kidney sellers is a difficult task. People cross the India-Nepal border easily and we cannot identify the kidney sellers, unlike women trafficking. The stories of kidney selling come up when there are disagreements in the cash negotiations between the sellers and brokers, only if they are not relatives to each other. It is not easy to know who is going to sell and who is taking them unless they are regular brokers.*

*(67 years old male, former policymaker at Nepal Planning Commission, KII)*

**Role of a broker.** The role of the brokers was central in kidney selling. The kidney brokers not only worked to find the sellers and buyers but were deeply involved in various steps of the illegal kidney trade. Brokers tended to identify potential sellers based on their network and contacts in the village. As most of the brokers were from the village; they knew the financial, psychological and social context of the community members. Brokers often exploited the financial difficulties faced by community members. After coaxing potential kidney sellers from the village, the brokers prepared all necessary paperwork that included a medical appointment for a regular check-up and fake documents to cross through the Nepal-India border. Finally, the brokers transported the kidney sellers to the venues where their kidneys were excised.

*One of the brokers tried to exploit the situation of a lady in the village. After the broker knew that her husband had an accident in Malaysia, he realised that she needed money and contacted her. Although she was convinced by the broker to sell her kidney, somehow the police stopped them and caught the broker who was sentenced to 3 years in Kavrepalanchok district prison.*

*(42 years old, female, legal worker, KII)*

The connection brokers had with people at different levels made their work easier including paperwork that required time and influence. The brokers were also good at forging official papers and negotiating with medical personnel and the buyers. In addition, the brokers were adept at convincing people, often persuading community members to sell kidneys by accentuating the financial gains in such a short period, and reassuring that selling one kidney would not be a problem for their health.

*I had sold my kidney because I thought it would give me a huge amount of money. I thought I would have a good life after selling it. The broker also told me that I will not have problems after selling because we have two and we can survive on one kidney.*

*(53 years old, Tamang, male, kidney seller, IDI)*

The brokers were flawless in acting as caretakers of the sellers. They took the potential sellers from the village to the venues where kidneys were excised. Based on their experience, the brokers were also cognizant of medical procedures around kidney transplantation such as blood matching. Once the blood matching was done, they would negotiate the price with potential buyers.

*If the sellers do not match with the receivers, brokers will try to find new receivers. But if it is matched, then the brokers start to negotiate with the buyers. Brokers are so smart they have*

*started to read the medical situation, psychology, economic conditions of the buyers and they increase the demand accordingly. These brokers have good observation skills, they are street-smart, and they can read and observe the patients.*

*(67 years old, male, a former policymaker at Nepal Planning Commission, KII)*

After selling the kidney, the brokers often influenced the sellers to identify other potential sellers in the village. This was the point where new brokers were borne and their network expanded

*Most of the sellers took their friends after selling their kidneys. They did not know about the health consequences therefore, they took their friends and relatives.*

*(42 years old, female, legal worker, KII)*

In many cases, the brokers were relatives of the kidney sellers in the village. Therefore, the sellers hesitated to report them to the legal authority, as one police officer confided to us:

*The brokers are their relatives. We tried to convince the victim to report their cases but they disagreed to file the case in the district police office.*

*(40 years old, male, local police officer, KII)*

## Discussion

This research identified that kidney selling is a complex social process arising from the blend of the vulnerability of community members and contextual factors which gave rise to a perfect scenario for brokers to demonstrate false hopes of alleviating the vulnerability and needs. Decision-making of community members in Hokse was affected by fellow kidney sellers with their stories of kidney selling and financial incentives. Remarkably, some of the victims of kidney selling later become brokers themselves and liaised in the system (Fig 3). One of the prominent reasons how kidney sellers later became brokers was because of their existing relationship and trust in their community where they saw an opportunity to generate an easy money. Below we discuss the study results in more detail.

Brokers are the central element of kidney trade in Hokse village. Previous reports highlight the role of poverty, and in particular the role of the earthquakes in pushing families into poverty which ultimately prompted them to sell their kidneys [33]. Nevertheless, attributing the kidney selling only to poverty sketches an incomplete story and requires a thorough exploration of social dynamics and mechanisms that underpin the kidney selling [18]. There are equally poor and/or poorer villages in Nepal that have not acceded to the kidney trade for livelihood. Previous research on organ and human trafficking have clearly shown that poverty is a fertile ground for increased vulnerability of the population who are the victims of human trafficking, prostitution, and organ trafficking [22, 34, 35]. In all of these activities, brokers have played a critical role and their major contribution in facilitating kidney trade should not be overlooked [36]. Similar findings are evident from the research conducted in Bangladesh, which details the role of brokers in transporting, identifying buyers, and negotiating with sellers and buyers [10].

Transplant surgeons are obliged to perform surgeries under the circumstances where the medical documentation and informed consent forms meet the standard criteria, although the informed consent forms are fake [37, 38]. Often the background related to these seemingly

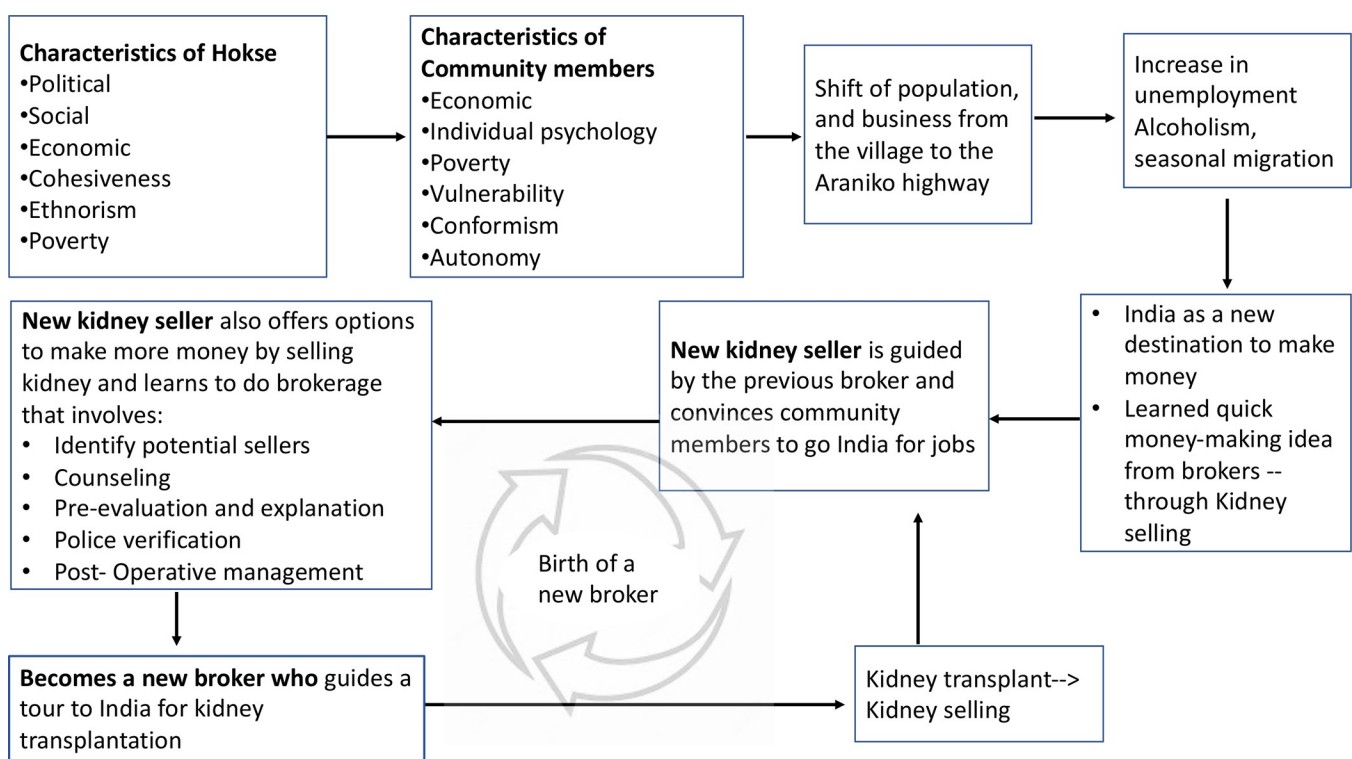

**Fig 3. This figure outlines how Hokse village became the kidney hub.** Along with the characteristics of village and community members, it shows a process of development of kidney brokers. Poverty and unemployment are exploited by a broker at first to offer them jobs in India. Further options for making more money including persuasion to sell kidneys are then implanted to the community member. As a result, the community member sells a kidney and returns to the village while remaining guided by the broker. In the pursuit of making easy money, the new kidney seller takes up a new role as a (new) broker and persuades community members to go to India for jobs and kidney selling.

perfect documentations are outside the scope of investigations for clinicians when they have met all the criteria. These practice of fake documentations were also evident in the other studies conducted in Bangladesh where the sellers were either prepared for the false kinship in Bangladesh or India [10]. Making up false kinship among kidney sellers and receivers was also reported in Nepal and explains a lot about the background on kidney trafficking documents [39].

One of the prominent reasons compelling community members to sell a kidney is their vulnerability due to poverty. Nonetheless, dumping all the reasons for kidney selling as 'poverty' is an understatement because there are poorer villages in Nepal than Hokse village which have not been victim to kidney selling. Indeed, studies have highlighted the role of poverty in kidney selling from low and middle-income countries such as Iran, Pakistan, India, the Philippines, and Bangladesh [10, 19, 40–44]. Poverty and illiteracy are highly prevalent among people of lower castes in Nepal [45]. Historically, the caste-based social hierarchy is dominated by higher castes who had more education, wealth and authority [46]. Over the years, such pre-domination remained ingrained deep into the societal dynamics and way of living leading to internalization of perceived inferiority, subjugation and dependence among lower castes [47]. Their tendency to look up to higher caste people also gradually offered them an impression that higher caste people's decision seems superior. The majority of lower caste people took up the decision because they saw that some of the higher caste people sold their kidneys. Such a decision-making process, and a tendency to follow higher caste people are reported from the region such as India and Bangladesh [36, 40, 48].

Drinking venues referred to as local pubs ('*bhattis*') which serve homemade rice/millet wines are a popular place to socialize. The combination of vulnerability, alcoholism, local '*bhattis*', and importantly the brokers are the perfect storm for kidney selling. The low paying non-technical jobs offered to the villagers and building relations with the buyers-sellers were performed in the local '*bhattis*' where the brokers introduce the topic of kidney selling to make more money. A similar pattern of false hope and deception for the poor slum dwellers were evident in the Philippines and the Indian Subcontinents [10, 37, 40, 41].

Although much has been described in the literature related to the roles and contributions of non-governmental aid organizations, their roles in Hokse village are unique particularly because of how their incentives to palliate the burden seem to foster dependence, false hopes and even promote the selling of kidneys among the community members. Euphemistic aids and their disruption of social dynamics are also highlighted by literature around temporary health camps in rural Nepal [49–51]. In the wider literature, global health and aid working organizations are criticized to perpetuate the white savior mentality, white supremacy and coloniality in Low and Middle-Income Countries in the forms of aids and humanitarian acts. Their acts have been incriminated to be negligible in terms of alleviating poverty, mitigating diseases and promoting development [52, 53]. In the case of Hokse village, temporary aids simply disrupted local livelihood. The community members understood that the kidney sellers get incentives from different organizations, therefore, they followed other neighbors to sell their kidneys.

Kidney trafficking is illegal according to the organ donation policy of Nepal. The policy highlights that *"No person shall operate an activity relating to organ transplantation for the purpose of the sale and purchase of an organ or similar other acts."* [54]. The sellers of the organ are mostly acknowledged for the deeds but the follow-up remains complex due to differences in nationality such as Nepalese selling their kidney to Indian nationalities and it is difficult to trace the sellers. Transplantation sites (being away from a person's own country), organ receiver's nationality and fake documents (paper tigers) add complexity to kidney trafficking. Specifically, the jurisdiction and legal liabilities become complex. Even more importantly, kidney trafficking is often reported to authorities only after serious disagreement or conflict between the organ sellers, receivers and brokers, mostly rooted in price negotiation and changes in deals at the last minute.

Legal specifications around organ donation also needs to revisit as organ donations are a sensitive topic. For instance, discussing the consent for organ donation from individuals/ patients in critical conditions (e.g., brain death) are difficult. Patients or their relatives can shun away from such discussions. The clarity in policies related to organ donation among such patients and their relatives can facilitate such discussions thereby preventing last-minute organ trafficking.

## Conclusion

Kidney selling was a result of a complex interaction of vulnerability and contextual factors at the individual and community level. Brokers exploited the vulnerabilities of the community members and were able to lure them into kidney selling. Brokers used several avenues for interaction with community members often through false hopes for better livelihood. The cycle of kidney selling is perpetuated further when victims themselves become new brokers. Although increasing support for livelihood and education is critical, regulatory measures are urgently needed to curb the kidney trade. First, regulatory authorities should mandate a thorough background check of a donor such as cross-checking with their family members, local authorities, and legal representatives. Second, facilitator can always be a broker disguising as a

family member and therefore needs to be suspected and explored for his/her potential role as a motivator. Third, cross-border collaboration between Nepal and India needs to scrutinize the legitimacy of consent process, documents, and potential involvement of a broker.

## Supporting information

**S1 File. Coreq checklist.**
(PDF)

**S2 File. Request for permission to publish content under CC-BY license.**
(PDF)

**S3 File. Codebook.**
(DOCX)

## Acknowledgments

We are very grateful to the participants who so generously took part in this study.

## Author Contributions

**Conceptualization:** Bijaya Shrestha, Manash Shrestha, Luechai Sringernyuang.

**Data curation:** Bijaya Shrestha.

**Formal analysis:** Bijaya Shrestha, Bipin Adhikari, Luechai Sringernyuang.

**Investigation:** Ankit Poudel, Binita Shrestha, Luechai Sringernyuang.

**Methodology:** Bijaya Shrestha, Bipin Adhikari, Manash Shrestha, Luechai Sringernyuang.

**Project administration:** Bijaya Shrestha.

**Supervision:** Bipin Adhikari.

**Visualization:** Dev Ram Sunuwar.

**Writing – original draft:** Bijaya Shrestha, Bipin Adhikari.

**Writing – review & editing:** Bijaya Shrestha, Bipin Adhikari, Manash Shrestha, Shiva Raj Mishra, Luechai Sringernyuang.

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
