## [Decision Letter · Decision Letter 0]

21 Apr 2022

PGPH-D-22-00320

‘The broker also told me that I will not have problems after selling because we have two and we can survive on one kidney’: findings from an ethnographic study of a village with one kidney in Central Nepal

Dear Dr. Shrestha,

Thank you for submitting your manuscript to PLOS Global Public Health. After careful consideration, we feel that it has merit but does not fully meet PLOS Global Public Health’s publication criteria as it currently stands. Therefore, we invite you to submit a revised version of the manuscript that addresses the points raised during the review process.

We look forward to receiving your revised manuscript.

Kind regards,

Peter Rohloff

Academic Editor

Journal Requirements:

1. In the online submission form, you indicated that "Data cannot be shared publicly because of the nature of the data being qualitative that contains personal quotes and clues to where the study occurred and can be potentially identifiable. However, data is available on request to the Academy for Data Sciences and Global Health, Kathmandu, Nepal complying with their data access policy.". All PLOS journals now require all data underlying the findings described in their manuscript to be freely available to other researchers, either 1. In a public repository, 2. Within the manuscript itself, or 3. Uploaded as supplementary information.

2. Please provide  separate figure files in .tif or .eps format only and remove any figures embedded in your manuscript file.  Please ensure that all files are under our size limit of 20MB.  

For more information about how to convert your figure files please see our guidelines: Once you've converted your files to .tif or .eps, please also make sure that your figures meet our format requirements

Additional Editor Comments (if provided):

I received two external reviews for this manuscript. Both reviewers were generally positive but have suggestions to improve the manuscript. Please review these suggestions and provide a detailed point by point response.

In addition, I have additional comments:

1. The data sharing statement doesn't adhere to the journal philosophy of open data access. Although it is true that qualitative data is often identifiable and can't be directly shared, the authors can prepare a more complete digest of the data for sharing. This should at least include the full code book used and a table of coding frequencies. This can be done without violating any confidentiality concerns.

2. The Discussion largely repeats the Results, leading to a lot of redundancy and not a lot of added value. The discussion should be distinct from the results, and it could be more synthetic and more engaged with international literature and with policy implications.

3. Although the background of the work is broadly ethnographic, the bulk of what is presented here is focused interviews with thematic coding. As such (and in line with journal policy) a checklist should be used to make sure all elements of the research design and methods are reported. Please use an EQUATOR recommended checklist to check and revise the paper (https://www.equator-network.org/reporting-guidelines-study-design/qualitative-research/)

Reviewers' comments:

Reviewer's Responses to Questions

**Comments to the Author**

1. Does this manuscript meet PLOS Global Public Health’s publication criteria? Is the manuscript technically sound, and do the data support the conclusions? The manuscript must describe methodologically and ethically rigorous research with conclusions that are appropriately drawn based on the data presented.

Reviewer #1: Yes

Reviewer #2: Yes

2. Has the statistical analysis been performed appropriately and rigorously?

Reviewer #1: N/A

Reviewer #2: N/A

3. Have the authors made all data underlying the findings in their manuscript fully available (please refer to the Data Availability Statement at the start of the manuscript PDF file)?

Reviewer #1: Yes

Reviewer #2: Yes

4. Is the manuscript presented in an intelligible fashion and written in standard English?

Reviewer #1: Yes

Reviewer #2: Yes

5. Review Comments to the Author

Reviewer #1: The manuscript is very well structured. The details that explain and describe the ethnographic approach/methodology are technically sound; and the theoretical framework references key publications on the topic of global organ trafficking in humans. The findings section includes a graphic illustration of the interaction between vulnerability and contextual factors that influence individual and community-level mindsets and experiences.

Minor edits suggested to the authors:

(a) Results section: Specify dates for fieldwork and data collection. These details are particularly relevant to better understand the significance of past events like the earthquake.

(b) Discussion section - overview: Consider adding a couple of sentences on the profile and mindset of the informant who was a seller and became a broker (in the results section), to better display the motivations/justifications behind liaising in the system in the discussion.

(c) Figure 3: Poverty due to earthquake is mentioned as relevant in the discussion, but it is not included as a village/community factor in Figure 3 (i.e., vulnerability due to natural disasters or adverse events).

(d) Conclusión: Specify a couple of examples of the most urgent “regulatory measured” that can contribute to curb the kidney market, based on the arguments presented in the discussion section.

*Please excuse any typos, as I am writing from the iPad.

Reviewer #2: The authors have done a good job in gathering and presenting the data on which the paper is based. The content of the discussion section follows logically from the authors' analysis of the factors behind the high rate of kidney selling in the village.

However, mysteriously missing from the paper is any reference to time. When was all this selling taking place? The authors say for decades. If for decades, why have the villagers not learned over time that the promises of the brokers that selling a kidney would not affect their health were not true? If about one third of villagers have sold a kidney, surely some have suffered medical complications. Has the rate of selling remained the same over the "decades" or gone up or down over time? Why? The authors indicate that bibliographic reference #28 provides more information regarding their methodology, but it is unclear what this reference is. What is the name of this journal?

Table 2 presents characteristics of the 10 kidney sellers. They range in age from the early 40s to 60. Presumably this is their age at the time of the study not at the time of their involvement in the trade. How old were they when they actually sold their kidney? What specifically did they need the money for? The gender ratio of men to women in the sample is 7:3. Is this ratio typical of the much larger population of sellers in the village? The authors state that altruistic kidney donors were not included in the study. But how do they know who was or was not an altruistic donor? All of the documentation that would-be donor/sellers fill out is to demonstrate that they are kin or altruistic donors since commercial sale is illegal. In fact, one of the most interesting quotes that the authors include is from a transplant surgeon who seemingly bemoans the fact that medical personnel are essentially compelled to carry out transplants when the paperwork is correctly (however falsely) filled out.

The authors make the important point that selling a kidney across borders complicates legal issues if something goes wrong with the transaction. This would certainly make cross border transaction more attractive to the brokers and buyers. Does that imply that selling occurs only or primarily as a cross-border phenomenon? Do the villagers take part in any commercial transactions within Nepal?

In their concluding sentence the authors state "regulatory measures are urgently needed to curb the kidney trade", but their paper has emphasized the total failure of the current regulatory regime to play any meaningful role in curbing the trade. What concrete regulatory changes do they see actually likely to have an impact on the organ trade?

Finally while generally well-written, there are places where copy editing would be useful. The authors should have a non-specialist reader go through the paper.

6. PLOS authors have the option to publish the peer review history of their article (what does this mean?). If published, this will include your full peer review and any attached files.

**Do you want your identity to be public for this peer review?** For information about this choice, including consent withdrawal, please see our Privacy Policy.

Reviewer #1: **Yes: **Ángel Eduardo del Valle Gonzalez

Reviewer #2: No

---

## [Decision Letter · Decision Letter 1]

7 Jul 2022

PGPH-D-22-00320R1

‘The broker also told me that I will not have problems after selling because we have two and we can survive on one kidney’: findings from an ethnographic study of a village with one kidney in Central Nepal

Dear Dr. Shrestha,

Thank you for submitting your manuscript to PLOS Global Public Health. After careful consideration, we feel that it has merit but does not fully meet PLOS Global Public Health’s publication criteria as it currently stands. Therefore, we invite you to submit a revised version of the manuscript that addresses the points raised during the review process.

We look forward to receiving your revised manuscript.

Kind regards,

Peter Rohloff

Academic Editor

Journal Requirements:

Additional Editor Comments (if provided):

The authors have responded comprehensively to the reviewer comments.

I appreciate the attention to detail with the checklist. The data availability statement. As requested in the last round, please submit as an supplementary file with your qualitative code book (list of codes) and coding frequencies, and then the paper will be ready for acceptance.

Reviewers' comments:

Reviewer's Responses to Questions

**Comments to the Author**

1. If the authors have adequately addressed your comments raised in a previous round of review and you feel that this manuscript is now acceptable for publication, you may indicate that here to bypass the “Comments to the Author” section, enter your conflict of interest statement in the “Confidential to Editor” section, and submit your "Accept" recommendation.

Reviewer #2: All comments have been addressed

2. Does this manuscript meet PLOS Global Public Health’s publication criteria? Is the manuscript technically sound, and do the data support the conclusions? The manuscript must describe methodologically and ethically rigorous research with conclusions that are appropriately drawn based on the data presented.

Reviewer #2: (No Response)

3. Has the statistical analysis been performed appropriately and rigorously?

Reviewer #2: (No Response)

4. Have the authors made all data underlying the findings in their manuscript fully available (please refer to the Data Availability Statement at the start of the manuscript PDF file)?

Reviewer #2: (No Response)

5. Is the manuscript presented in an intelligible fashion and written in standard English?

Reviewer #2: (No Response)

6. Review Comments to the Author

Reviewer #2: (No Response)

7. PLOS authors have the option to publish the peer review history of their article (what does this mean?). If published, this will include your full peer review and any attached files.

**Do you want your identity to be public for this peer review?** For information about this choice, including consent withdrawal, please see our Privacy Policy.

Reviewer #2: No

---

## [Decision Letter · Decision Letter 2]

22 Sep 2022

‘The broker also told me that I will not have problems after selling because we have two and we can survive on one kidney’: findings from an ethnographic study of a village with one kidney in Central Nepal

PGPH-D-22-00320R2

Dear Mr Shrestha,

We are pleased to inform you that your manuscript '‘The broker also told me that I will not have problems after selling because we have two and we can survive on one kidney’: findings from an ethnographic study of a village with one kidney in Central Nepal' has been provisionally accepted for publication in PLOS Global Public Health.

Best regards,

Ajay Bailey, Ph. D.

Academic Editor

The reviewer is accepts the revisions and I am happy to accept the paper.

Reviewer Comments (if any, and for reference):

Reviewer's Responses to Questions

**Comments to the Author**

1. If the authors have adequately addressed your comments raised in a previous round of review and you feel that this manuscript is now acceptable for publication, you may indicate that here to bypass the “Comments to the Author” section, enter your conflict of interest statement in the “Confidential to Editor” section, and submit your "Accept" recommendation.

Reviewer #2: All comments have been addressed

2. Does this manuscript meet PLOS Global Public Health’s publication criteria? Is the manuscript technically sound, and do the data support the conclusions? The manuscript must describe methodologically and ethically rigorous research with conclusions that are appropriately drawn based on the data presented.

Reviewer #2: (No Response)

3. Has the statistical analysis been performed appropriately and rigorously?

Reviewer #2: (No Response)

4. Have the authors made all data underlying the findings in their manuscript fully available (please refer to the Data Availability Statement at the start of the manuscript PDF file)?

Reviewer #2: (No Response)

5. Is the manuscript presented in an intelligible fashion and written in standard English?

Reviewer #2: (No Response)

6. Review Comments to the Author

Reviewer #2: The authors successfully addressed all my concerns.

7. PLOS authors have the option to publish the peer review history of their article (what does this mean?). If published, this will include your full peer review and any attached files.

**Do you want your identity to be public for this peer review?** For information about this choice, including consent withdrawal, please see our Privacy Policy.

Reviewer #2: No
